# PhytoAFP: In Silico Approaches for Designing Plant-Derived Antifungal Peptides

**DOI:** 10.3390/antibiotics10070815

**Published:** 2021-07-05

**Authors:** Atul Tyagi, Sudeep Roy, Sanjay Singh, Manoj Semwal, Ajit K. Shasany, Ashok Sharma, Ivo Provazník

**Affiliations:** 1Department of Biomedical Engineering, Faculty of Electrical Engineering and Communication, Brno University of Technology, Technicka 12, 61600 Brno, Czech Republic; tyagi@vut.cz (A.T.); roy@vut.cz (S.R.); 2Biotechnology Division, CSIR—Central Institute of Medicinal and Aromatic Plants, P.O.—CIMAP, Near Kukrail Picnic Spot, Lucknow 226 015, Uttar Pradesh, India; sanjay.singh@cimap.res.in (S.S.); m.semwal@cimap.res.in (M.S.); ak.shasany@cimap.res.in (A.K.S.); sharmaas58@gmail.com (A.S.); 3Department of Physiology, Faculty of Medicine, Masaryk University Brno, Kamenice 5, 62500 Brno, Czech Republic

**Keywords:** plant defensins, innate immunity, host defense peptides, antimicrobial peptides

## Abstract

Emerging infectious diseases (EID) are serious problems caused by fungi in humans and plant species. They are a severe threat to food security worldwide. In our current work, we have developed a support vector machine (SVM)-based model that attempts to design and predict therapeutic plant-derived antifungal peptides (PhytoAFP). The residue composition analysis shows the preference of C, G, K, R, and S amino acids. Position preference analysis shows that residues G, K, R, and A dominate the N-terminal. Similarly, residues N, S, C, and G prefer the C-terminal. Motif analysis reveals the presence of motifs like NYVF, NYVFP, YVFP, NYVFPA, and VFPA. We have developed two models using various input functions such as mono-, di-, and tripeptide composition, as well as binary, hybrid, and physiochemical properties, based on methods that are applied to the main data set. The TPC-based monopeptide composition model achieved more accuracy, 94.4%, with a Matthews correlation coefficient (MCC) of 0.89. Correspondingly, the second-best model based on dipeptides achieved an accuracy of 94.28% under the MCC 0.89 of the training dataset.

## 1. Introduction

Acquired resistance to fungal infections is rapidly becoming a major medical problem. Opportunistic fungal infections pose a treatment challenge, especially in high-risk immune-compromised patients receiving AIDS, cancer, and transplantation treatment [1]. Infectious diseases that cause death pose a significant challenge for public health. Handling fungal infections in an immune-deficient sufferer is quite challenging. In addition, drug efficacy, toxicity, and the development of resistance to antibiotics are also significant challenges [2].

Fungi cause several diseases in plant crops, resulting in losses and decreased productivity, quality, and welfare. Plant disease treatment mainly depends on chemical pesticides, which are sharply restricted (American Phytopathological Society) [3,4]. With the rising issue of antibiotic-resistant pathogens [5,6,7] and the decreasing number of antibiotics capable of combating these infections, coupled with pharmaceutical companies’ reluctance to invest in infectious disease research [8,9], the demand for new antifungal drugs has never been so urgent [10]. After the 1980s, antimicrobial peptides have been lauded for their potential as novel antibiotics to discover antimicrobial peptides [11].

This also depends on different factors, including structural and physicochemical composition information from peptides and genetically characteristic fungal host species, suggesting resistance or susceptibility to peptides [12]. These antifungal peptides have been the object of attention for many years as plant protection molecules [13]. PhytoAFP are the natural drug sources that stimulate plant health, thereby obtaining new output in crop protection research that complies with new regulations [10,14,15,16]. Most antimicrobial peptides come from different plant species and play an essential role in the defense of plants against fungi [10]. These antimicrobial peptides that exhibit antifungal activity are called antifungal peptides. Several potential therapeutic antimicrobial peptides have been reported, but only a few have been experimentally annotated [17]. Therefore, there is a considerable gap between new and expected therapeutic antifungal peptides. Many protein peptides against fungi have been discovered from different sources such as plants, animals, amphibians, and humans [18].Their broad-spectrum antibacterial activity and the selectivity of bacteria to eukaryotic cells make them attractive candidates for novel pharmaceutical compounds. Plant-origin antifungal peptides (PhytoAFP) might cause morphological, physiologic, and molecular disruption and kill the fungal cell in some cases (Figure 1).

Due to a higher stage of development of peptide-based therapeutics, pharmaceutical companies focus more on peptide-based drugs [19]. A large number of calculation tools, such as AntiCP [20], ToxinPred [21], TumorHPD [22], AVPpred [23] and CellPPD [24] have been developed to predict and design anticancer peptides, toxic peptides, plant antifungal peptides, antiviral peptides, and cell-penetrating peptides, respectively. We extracted 510 experimentally validated, manually curated peptides from the PlantAFP database for SVM learning. We have not included modified or non-natural amino acids in our studies. We collected 510 random peptides from the UniProt database and considered them as a negative dataset. Thus, the PhytoAFP main dataset contains 510 PhytoAFP and 510 Non-PhytoAFP. In this case, positive refers to antifungal peptides, whilst negative refers to non-antifungal peptides. The architecture of our current work is provided in Figure 2.

We have developed a PhytoAFP web prediction server where a user can predict and design therapeutic antifungal peptides in the current study. It also provides different modules to generate possible mutants of a given peptide. The treatment of fungal infections in immunodeficient patients, besides toxicity, drug efficacy, and resistance to antibiotics, are also significant challenges. Therefore, the scientific community is more interested in the emerging field of therapeutic plant-derived antifungal peptides. As far as we know, this is the first method developed to predict and design potential plant-derived antifungal peptides. Our method may help researchers find and design better peptide-based antifungal drugs.

## 2. Results

### 2.1. Frequency of Occurrence of All Twenty Natural Amino Acids

The average percent composition of amino acids in PhytoAFP peptides and Non-PhytoAFP peptides have been calculated and compared to determine the frequency of occurrence of all twenty natural amino acids. We calculated the average percent composition of amino acids in PhytoAFP peptides and Non-PhytoAFP peptides to compare them in order to determine the frequency of occurrence of all twenty natural amino acids. We have calculated the percent composition of both N- and C-terminus. Cys, Glu, Gly, Lys, Gln, Arg, Ser, and Tyr are more abundant in PhytoAFP than the Non-PhytoAFP peptide dataset. To understand whether certain types of amino acids are dominant in PhytoAFP, the average percentage composition of amino acids in PhytoAFP and Non-PhytoAFP has been calculated and compared (Figure 3). We have observed that certain residues such as Cys, Phe, Gly, His, Lys, Asn, Pro, Arg, and Tyr are more abundant in PhytoAFP. Therefore, we are also interested in knowing whether certain types of residues dominated at N-terminus and C-terminus in PhytoAFP. Consequently, we have determined the overall percentage of residue preference in the N and C termini (residue amino acid composition) to solve this problem. However, we have not distinguished the apparent distinction of amino acids from the total residue composition of PhytoAFP (Figure 3).

### 2.2. Analysis of PhytoAFP Based on Residue Preference Using Two Sample Logos

Two sample logos are calculated to understand the residue preference (N-terminus and C-terminus) at the end. The first ten residues are used to generate the logo from the two ends. As shown in Figure 4A at the N-terminus, residue A is in the first position, A is in the second position, and C is dominant in the third position. In addition to these residues, G is also preferred in other positions. The C and R residues are highly preferred at the C-terminus compared to other amino acids (Figure 4B).

### 2.3. Motif Analysis Using MERCI

The MERCI tool identified motifs and familiar patterns in PhytoAFP and Non-PhytoAFP within the query sequences [25].We used the default standard when running the program. These motifs are extracted from PhytoAFP (positive) and Non-PhytoAFP (negative) datasets. The MERCI motif program allows the comparison of positive and negative peptides to extract motifs. To understand the subject, we used a two-step strategy. The two-step strategy shows that there are motifs in PhytoAFP and Non-PhytoAFP. Firstly, in the two-step strategy method, PhytoAFP was used as positive input and Non-PhytoAFP used as negative input. In the second step, we reversed the order of input where Non-PhytoAFP and PhytoAFP were used as positive and negative input, respectively. Finally, we get the results of many numbers of motifs present in PhytoAFP and Non-PhytoAFP, which can be further used in scanning peptides for antifungal-specific motifs. The sequence analysis of the main dataset shows that there are 50 exclusive motifs in the positive datasets and 49 exclusive motifs in the negative data set (Appendix A).

### 2.4. Analysis of PhytoAFP Using Composition-Based Models

In the compositional analysis of PhytoAFP, it was noticed that specific residues are more dominant. This indicates that PhytoAFP and Non-PhytoAFP may differ in their amino acid composition methods. Model-based observations of amino-acid-based components (AAC), dipeptide components (DPC), and tripeptide components (TPC) were developed using SVM models on the main data set. The performance of the AAC-based (monopeptide) SVM model is shown in Table 1. The model developed on the PhytoAFP main dataset (TPC-based model) achieved a maximum accuracy of 94.4% with an MCC and AUC (area under the curve) of 0.89 and 0.98, respectively. Similarly, the dipeptide composition-based model achieved an accuracy of 94.28% with MCC 0.89 for the training dataset (Table 1). Correspondingly, the subsets NT5, CT5, NTCT5, NT10, CT10, NTCT10, and performances of these models, are summarized in Table 2. For example, based on five-fold cross-validation, a model developed with the NTCT10 dataset achieved a higher accuracy of 91.71% with MCC 0.84 and AUC 0.96, respectively (Table 2). To test the performance of the models, a five-fold cross-validation technique is used. Four sets were used for training, and the remaining one was used for testing. The whole process was repeated five times. Threshold-dependent parameters based on accuracy and the MCC measurements revealed that di- and tripeptide SVM models were selected as the best models. The performance of the selected models was evaluated on an independent dataset. To validate our model, we evaluated the performances of our best models (di-and tripeptide composition) on an independent dataset. For the pipeline to develop a classification method, we used 402 plant antifungal peptides and 402 non-plant antifungal peptides for training and testing (training data set). We use the remaining 100 plant antifungal peptides as a validation data set, that is, for external cross-validation (Figure 2).

### 2.5. Analysis of PhytoAFP Using the Binary Pattern-Based Method

Using binary patterns as input features, SVM models have been generated for window sizes of 5 and 10 residues. To this end, 5 and 10 residues have been extracted from the N-terminal and C-terminal of each data set, respectively. A window pattern of 5 and 10 residues has been used for the N and C termini. We have removed all the peptides having a length of less than 5 and 10, respectively. We achieved a maximum accuracy of 92.71% in the NTCT15-BIN binary pattern in the five-fold cross-validation method with MCC 0.86 and ROC 0.97171, respectively (Table 3).

### 2.6. Performance on the Independent Dataset

To validate our models, we evaluated the performances of our two best models on an independent dataset. In the tripeptide composition-based model, we found a maximum accuracy of 94.40% and MCC of 0.89 for the training dataset and accuracy of 90.05% and MCC value 80% for the validation dataset. Similarly, the dipeptide composition-based model achieved an accuracy of 94.28% with MCC of 0.89 for the training dataset and an accuracy of 91% and MCC value of 0.82% for the validation dataset, suggesting that our models are helpful for further studies.

### 2.7. ROC Plot

To perform an independent threshold evaluation of our models, we have generated ROC (Receiver Operating Characteristic) curves for all models. The R studio software package of the R language is used to create ROC plots with the area under the curve (AUC) [19]. The AUC provides values that are important to consider when checking the performance of the method. The value suggests that dipeptide- and tripeptide-based approaches performed better in our studies (Figure 5). 

## 3. Discussion

In the current work, we applied a systematic approach to analyses to predict plant-derived antifungal peptides. As observed in the compositional analysis, PhytoAFP peptides have preference fora specific type of residues Cys, Phe, Gly, His, Lys, Asn, Pro, Arg, and Tyr. The compositional analysis provides essential information for experimental biologists to design antifungal peptides. The main challenge is to develop an accurate method to predict PhytoAFP peptides. We calculated various features, including amino acid composition; dipeptides, tripeptides, split amino acids, hybrid and binary method as input features.

The composition-based method gave better results, and the monopeptide-, dipeptide-, tripeptide-based methods as input vectors also performed well. In all data sets, the monopeptide-, dipeptide-, tripeptide-based methods gave the maximum ROCs of 0.97%, 0.97%, and 0.98%, respectively (Table 1). Thus, because di-and tripeptide composition-based methods performed well in the SVM, we have implemented them as models for our server. The computational approaches provided by our group open new avenues for antifungal peptide prediction and for the design of therapeutics in the drug development process globally. PhytoAFP peptides have variations in length, and most peptides lie in the range between 5 and 10 residues (Table 2). We have developed two types of datasets based on peptides of variable lengths and another with peptides within the range of 5–10 residues. SVM is entirely independent of sequence similarity. The only requirement for the SVM is fixed length as an input pattern. This directly allows us to work and analyze those peptides that are less similar. We distinguished PhytoAFP peptides from Non-PhytoAFP peptides using mono amino acid as an input vector. From monopeptide (APC) and binary profile composition, we derived N-terminal residue and C-terminal residue of window sizes 5 and 10, which allowed us to develop an SVM model of fixed peptide length (Table 3). 

The designed server is very convenient for researchers working in plant-based development of new antifungal peptides against emerging infectious diseases. PhytoAFP peptides may be helpful in this regard, but the challenge remains the availability and effectiveness of these peptides. The significance of the developed peptides will depend entirely on further validation studies, including gene expression analysis RNA-seq, MS identification, protein localization, and ease of peptide synthesis. The complete architecture of the PhytoAFP is provided in (Figure 2).

### 3.1. PhytoAFP Web Server

The PhytoAFP web server provides a different module for generating possible single mutants or analogs of the specified peptides and predicting their plant-derived antifungal peptide activity and physicochemical properties, such as charge hydrophobicity and pI. Thus, this server allows users to design PhytoAFP and its mutants with different physicochemical properties. There are three major modules of the PhytoAFP web server, and these are as follows.

#### 3.1.1. Peptide Design

This module allows users to generate all possible single mutant analogs of their peptides and predict whether the analogs will have antifungal properties or not. This peptide design module will enable users to submit and design single plant antifungal peptides. It will generate all the possible mutants of the given peptide and predict their antifungal activity along with all the essential physicochemical properties, such as hydrophobicity, charge, and pI, selected by the user in the display of the SVM-based option. The SVM-based method predicts the antifungal efficiency based on SVM scores, which use the monopeptide composition of the peptide as input. The user must choose the SVM threshold from which plant antifungal peptide and non-plant antifungal peptide will be classified.

#### 3.1.2. Multiple Peptides

The multiple peptide module allows users to predict the antifungal properties from multiple peptide entries to be designed by the SVM method. The rest of the functional tabs are the same as the peptide design module discussed above.

#### 3.1.3. Protein Scan

This module generates all the possible overlapping peptides and their single mutant analogs of the protein submitted by the user. It also predicts whether an overlapping peptide/analog is PhytoAFP or Non-PhytoAFP. 

Here, the user can extract a protein sequence to search plant antifungal peptide sequences within the protein sequence using the SVM method. Users must select the length of the fragmented peptide so that only peptides with a specific length will be generated.

### 3.2. Sequence Download Webpage

The sequence download webpage has been embedded in the PhytoAFP server, where users can download the data set used for the current study.

## 4. Materials and Methods

Plant-derived antifungal peptides were extracted from the PlantAFP database [10], which provides comprehensive information on a wide range of PhytoAFP. The PlantAFP database consists of ~2585 peptide entries, corresponding to 510 unique antifungal peptides. All these peptides are unique and are considered positive examples. There are few experimentally proven Non-PhytoAFP, since there are very few experimentally verified Non-PhytoAFP. We extracted 510 random peptides from the Swissport protein and regarded them as negative examples (Non-PhytoAFP). Thus, the main dataset contains 510 PhytoAFP and 510 Non-PhytoAFP. We used amino acid composition, dipeptide composition, tripeptide composition, binary profile composition, split acid composition, or a hybrid approach, as well as physicochemical properties for feature selection in this approach. We used various features as SVM inputs for the prediction of the PhytoAFP dataset in the present study. We used the following procedures for the development of SVM models.

### 4.1. Cross-Validation Technique 

Cross-validation is necessary for any prediction method. In our study, the five-fold and ten-fold cross-validation technique has been utilized to evaluate working models’ performance. Cross-validation is a valuable and reliable means for testing the prediction ability of methods. The PhytoAFP datasets were randomly split into two parts: (1) the training dataset, which constitutes 80% of the data and (2) the validation dataset, with the remaining 20% of the data. For internal validation, we developed prediction models and evaluated them using a five-fold cross-validation technique. In five-fold cross-validation technique, the sequence is randomly divided into five datasets, of which any four data sets are used for training and the fifth data set is used for testing purposes. In order to use each of the five data sets for testing, we repeat this process five times. The result is calculated by averaging the performance of all five groups. We evaluated the model’s performance using the training data set on the validation data set for external validation. Evaluation of different significant SVM modules’ performances has been applied by calculating the accuracy and Matthews correlation coefficient (MCC).

### 4.2. Measuring Parameters

The performance of each model developed in our research is calculated by using threshold-related parameters and threshold-independent parameters. For the threshold-dependent parameters, we utilized specificity, sensitivity, MCC, and overall accuracy using the equations provided below (Section 4.3). TP is a positive example of accurate prediction, and TN is a negative example of accurate prediction. Correspondingly, FP was incorrectly predicted as positive, and FN was incorrectly predicted as negative. The predictions and prediction categories of any method are divided into four categories from the original data set. False positive is the result that indicates a model condition that exists when it does not. Similarly, false negative is the result that indicates a model condition that does not exist when it does. We also calculated the AUC-ROC of the final model to estimate the model’s performance using threshold-independent parameters (Figure 5).

### 4.3. Threshold-Dependent Measures

The predicted performance calculated from different thresholds or cut-offs and four other parameters have been used to estimate the performance, as described below.

**Accuracy (ACC):** The prediction accuracy is the number of accurate predictions in the total prediction (TP + TN). When the data set is in equilibrium (positive and negative datasets are equal), evaluating any method’s performance is a reasonable estimate.
Accuracy = (TP+TN)/(TP+FP+TN+FN) ×100

**Sensitivity (Sen):** This measures the percentage of correct predictions (TP) in all positive examples. It is also known as percent coverage or recall.
Sensitivity = TP/(TP+FN) ×10

**Specificity or precision (Spe):** This measures the percentage of accurate predictions (TN) in all negative examples. Higher specificity means that there are almost all negative cases within the expected results. However, at the same time, some positive examples will also be predicted as positive.
Specificity= TN/(TN+FP) ×100

**Positive predictive value (PPV):** This is the percentage of correctly predicted positive instances (TP) in the total number of positive predictions (TP + FP).

**Matthews correlation coefficient (MCC):** One of the best performance measures, which accounts for both over- and under-prediction, is the Matthews correlation coefficient. It is the best measure of performance if the dataset is imbalanced. The value of MCC ranges from –1 (perfectly non-anti-correlated) to 1 (perfectly correlated). If there are no relationships between the observed (predicted) values and the actual values, the correlation coefficient is 0 or very low. As the relation between the actual values and predicted value increases, the same is true for the correlation coefficient.
MCC=((TP)(TN)−(FP)(FN))/√([TP+FP][TP+FN][TN+FP][TN+FN])

Threshold-independent measures: The main limitation of threshold-related measurements is that they can calculate a specific threshold at most, so an alternative method that is independent of the threshold is needed. The Receiver Operating Characteristic (ROC) is an independent threshold measure that was introduced as a signal processing technique. An ROC graph shows all true positive scores (sensitivity) on the y-axis relative to the false-positive scores (specificity) of all thresholds on the x-axis. The curve is drawn on two points (0, 0, and 1, 1), and0, 0 is where no positive cases can be found, so it always indicates that negative cases are correct, and all positive cases are wrong. The R studio software package of the R language has been used to determine the prediction method’s ROC value. It calculates an ROC curve and expands the area under the curve.

## 5. Conclusions

In response to the demand for new drug delivery systems, research in antifungal peptides for emerging infectious diseases has grown rapidly. The designing of the PhytoAFP webserver is an effort to design and predict highly effective antifungal peptides. It also allows and helps to find newer PhytoAFP analogs faster and more efficiently. We hope that establishing such a method will speed up identifying improved and efficacious PhytoAFP in the future. PhytoAFP is a unique web server where users can predict, design, and derive activities for antifungal peptides and can be accessed at http://bioinformatics.cimap.res.in/sharma/PhytoAFP (accessed on 1 July 2021).

## Figures and Tables

**Figure 1 antibiotics-10-00815-f001:**
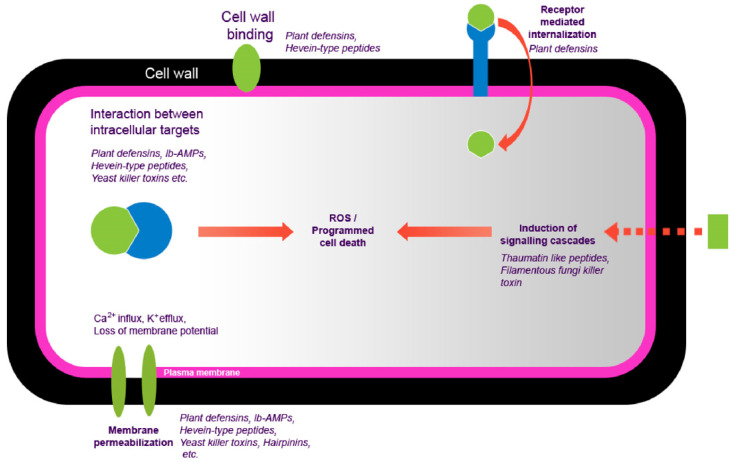
The mechanism of action of plant-derived antifungal peptides.

**Figure 2 antibiotics-10-00815-f002:**
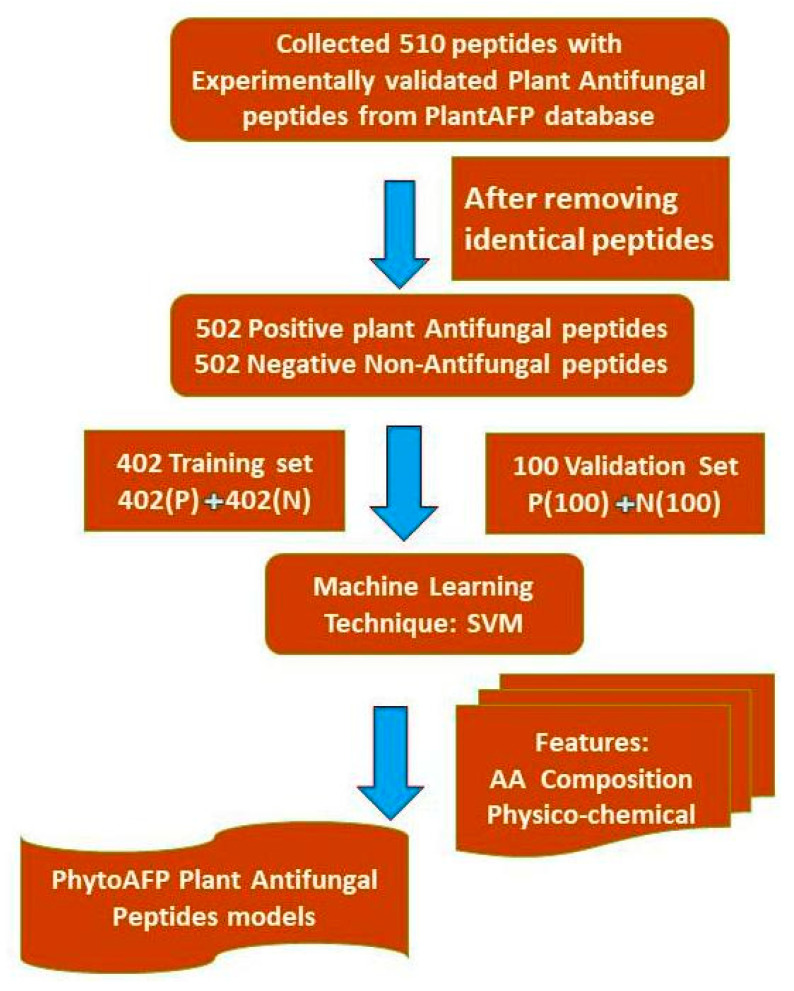
Architecture of PhytoAFP.

**Figure 3 antibiotics-10-00815-f003:**
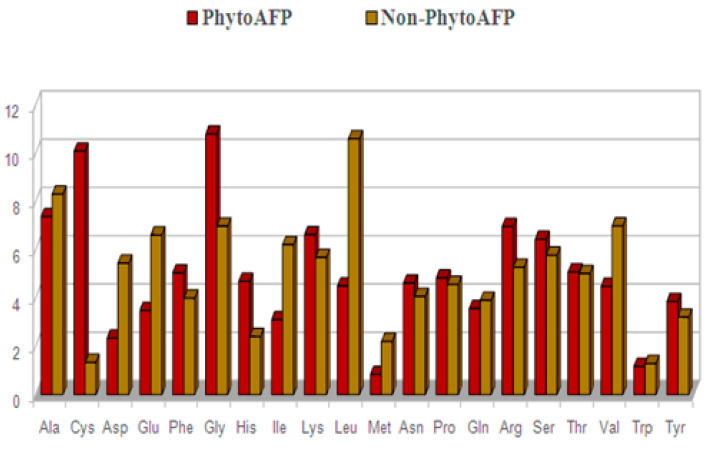
Amino acid composition of PhytoAFP and Non-PhytoAFP.

**Figure 4 antibiotics-10-00815-f004:**
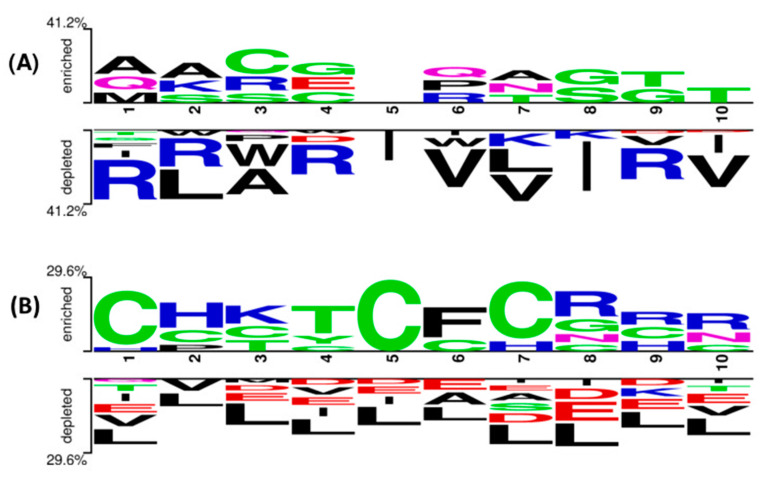
Two sample logos illustrate the position preference of the first 10 residues present in PhytoAFP and Non-PhytoAFP at (**A**) N-terminus and (**B**) C-terminus, respectively.

**Figure 5 antibiotics-10-00815-f005:**
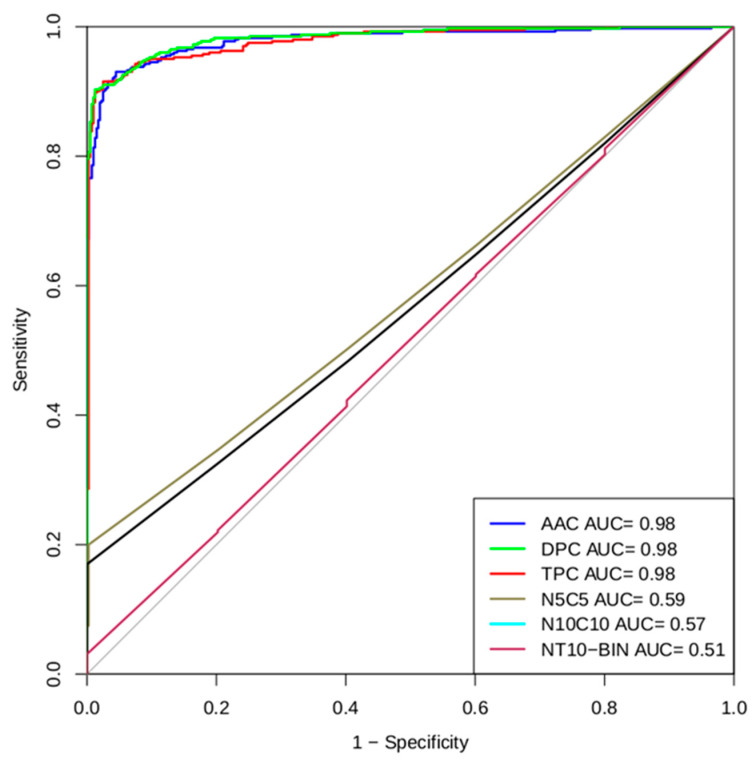
Multi-ROC plot shows a performance of best models developed using AAC, DPC, TPC, N5C5, N10C10, and NT10-Bin peptide compositions in five-fold cross-validation.

**Table 1 antibiotics-10-00815-t001:** Performances of SVM models developed using mono-, di-, and tripeptide acid composition of peptides using five-fold cross-validation.

Main Dataset						
Input Vector	SVM Parameters	Sensitivity	Specificity	Accuracy	MCC	ROC
Monopeptide	g:0.005 c:9 j:1	93.03	95.52	94.28	0.89	0.97947
Dipeptide	g:0.0005 c:7 j:2	91.54	97.01	94.28	0.89	0.97868
Tripeptide	g:0.0005 c:7 j:2	90.05	98.76	94.4	0.89	0.98361

**Table 2 antibiotics-10-00815-t002:** Performance of SVM models developed using split acid composition using five-fold cross-validation.

Main Dataset						
Input Vector	SVM Parameters	Sensitivity	Specificity	Accuracy	MCC	ROC
NT5	g:0.0005 c:3 j:2	85.75	85.82	85.79	0.72	0.90951
CT5	g:0.0001 c:9 j:1	74.14	92.54	84.37	0.69	0.88318
NT10	g:0.005 c:1 j:2	82.23	93.42	87.95	0.76	0.93393
CT10	g:0.0005 c:4 j:1	94.43	88.34	0.77	0.92903	94.43
NTCT5	g:0.0005 c:1 j:3	88.25	92.29	90.27	0.81	0.95398
NTCT10	g:0.001 c:2 j:1	86.21	96.96	91.71	0.84	0.96315

**Table 3 antibiotics-10-00815-t003:** Performances of SVM models developed using a binary profile-based method using five-fold cross-validation.

Main Dataset						
Input Vector	SVM Parameters	Sensitivity	Specificity	Accuracy	MCC	ROC
NT5-BIN	g:0.5 c:1 j:1	78.2	98.01	88.14	0.78	0.93761
CT5-BIN	g:0.05 c:3 j:3	95.87	80.85	87.45	0.76	0.96817
NT10-BIN	g:0.1 c:2 j:1	98.67	60	90.89	0.7	0.92579
CT10-BIN	g:0.05 c:6 j:1	80.33	93.42	87.12	0.75	0.91071
NT15-BIN	g:0.1 c:1 j:2	83.95	95.74	90.29	0.81	0.9383
CT15-BIN	g:0.1 c:1 j:2	76.43	98.4	88.41	0.78	0.90965
NTCT5-BIN	g:0.1 c:5 j:1	87.22	83.58	85.39	0.71	0.92252
NTCT10-BIN	g:0.05 c:3 j:1	99.47	47.37	88.98	0.63	0.89465
NTCT15-BIN	g:0.05 c:2 j:1	86.11	98.4	92.71	0.86	0.97171

## Data Availability

Data is contained within the article.

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
