# Peer review of "PhytoAFP: In Silico Approaches for Designing Plant-Derived Antifungal Peptides"

_antibiotics, 2021, doi:10.3390/antibiotics10070815_

Round 1

Reviewer 1 Report

Infectious diseases are a major threat to public health. Fungal infections represent an important role in this threat. So, the relevance of this work is high. The need for new drugs to fight fungal infections is important.

Line 43 – 46: Hence, keeping problems in mind, scientists are more attracted to the emerging rea of therapeutic antifungal peptides with innate immunity activities. Thus, keeping the issues mentioned above in mind, scientists are more attracted to the emerging therapeutic antifungal peptides with innate immunity activities.

The information in these lines is repeated.

Line 48 – 51: Plant disease authority mainly depends on chemical pesticides, which are sharply restricted (American Phytopathological Society) [3, 4]. Plant-fungal diseases are most dependent on chemical pesticides, which are currently highly regulated as per essential regulatory requirements[5].

These two sentences are describing the same information. Must be reformulated. It is presented results in the abstract that are not discussed in the main document.

The discussion is reduced. I think it would be beneficial of a more comprehensive discussion. A case study to complement the work described would be beneficial.

Author Response

PhytoAFP webserver comments point-by-point response to Ref.: Ms. No. 1241099

Thank you very much for yourcommentsand suggestions for helping us improve the manuscript. We have incorporated all your answers to your comments and suggestions into our revised manuscript.We have rewritten most of the content under different sections, keeping the original subject matter intact in our manuscript. Changes are highlighted in blue color. We are hereby addressng the responses on a point-by-point basis.

Reviewer #1: PhytoAFP webserver comment:

Infectious diseases are a major threat to public health. Fungal infections represent an important role in this threat. So, the relevance of this work is high. The need for new drugs to fight fungal infections is important.

Point #1: -- Line 48 – 51: Plant disease authority mainly depends on chemical pesticides, which are sharply restricted (American Phytopathological Society) [3, 4]. Plant-fungal diseases are most dependent on chemical pesticides, which are currently highly regulated as per essential regulatory requirements [5]. These two sentences are describing the same information. Must be reformulated. It is presented results in the abstract that are not discussed in the main document.

Response: We have reformulated the sentence provided between Line 48 – 51 in our revised manuscript. We have incorporated our findings in both abstract and main document as suggested by the reviewer [see line number 59-62].

Point #2: --The discussion is reduced. I think it would be beneficial of a more comprehensive discussion. A case study to complement the work described would be beneficial.

Response:  We agree with the reviewer that there is a need to improve and provide a comprehensive “Discussion” section. Therefore, as suggested, we have modified our manuscript and added the comprehensive discussion under the “Discussion” section. We have tried to discuss our findings based on the results obtained to the best possible extent. Since limited work has been done in this field hence, we are not able to provide a case study. The importance of work is immense. The current work will largely benefit the agriculture sector [see line number 163-237].

Reviewer 2 Report

Manuscript Title: PhytoAFP: In silico approaches for designing plant-derived antifungal peptides

My review is as follows:

Emerging infectious diseases caused by fungi in humans and plant species have posed problems to food security and consequently to human health.  The authors aimed to build a platform to collect the physical and chemical properties of the plant-derived anti-fungal peptides including peptide sequence, peptide name, and antifungal activity, etc.  The authors tried to describe the logics of their analysis algorithm in the platform and proposed to rationally design the therapeutic anti-fungal peptides in the manuscript.

The comments and suggestions are listed as followed.

  1. a very similar platform, PlantAFP was available and published two years ago (http://bioinformatics.cimap.res.in/sharma/PlantAFP). The authors should carefully describe the differences between these two platforms.
  2. Some of the pulldowns in the PhytoAFP platform (http://bioinformatics.cimap.res.in/sharma/PhytoAFP) was not available, especially no database can be accessed.
  3. In order to well predict the therapeutic anti-fungal peptides, the authors should carefully choose the training database with the known therapeutic anti-fungal peptides. The authors should carefully describe their criterions of choosing the sequences in the training database. 
  4. The authors should also carefully describe the logics of their algorithm and show the efficacy of their prediction. In other words, the test database should be built and the predicted anti-fungal “leads” should be proved experimentally.
  5. The manuscript should be carefully proof-read.

Author Response

PhytoAFP webserver comments point-by-point response to Ref.: Ms. No. 1241099

Thank you very much for yourcomments, and suggestions for helping us improve the manuscript. We have incorporated all your answers to your comments and suggestions into our revised manuscript.We have rewritten most of the content under different sections, keeping the original subject matter intact in our manuscript. Changes are highlighted in blue color. We are hereby addressing the responses on a point-by-point basis.

Reviewer #2: PhytoAFP webserver comment:

Emerging infectious diseases caused by fungi in humans and plant species have posed problems to food security and consequently to human health.  The authors aimed to build a platform to collect the physical and chemical properties of the plant-derived anti-fungal peptides including peptide sequence, peptide name, and antifungal activity, etc.  The authors tried to describe the logics of their analysis algorithm in the platform and proposed to rationally design the therapeutic anti-fungal peptides in the manuscript.

Point #1: -- a very similar platform, PlantAFP was available and published two years ago (http://bioinformatics.cimap.res.in/sharma/PlantAFP). The authors should carefully describe the differences between these two platforms.

Response: “PlantAFP” which the reviewer has indicated relating it to similar working platform like PhytoAFP (Current work) that was published 2 years back is a database not a webserver. The previous work is an effort from our group under which we designed a database that have experimentally validated plant specific activity-based manually curated anti-fungal peptide data. We incorporated data related to plant species, family, tissue localization of plant antifungal peptides in PlantAFP database.

In our current manuscript, we have developed a webserver named PhytoAFP, for high-precision prediction and design of antifungal peptides. Users can submit peptides to the server and predict whether a given peptide possess anti-fungal properties or not. We had provided the option for the users to generate mutants of a query peptide(s). Another wonderful feature that has been incorporated allows the user to identify minimum mutations required to increase the potency of the input peptide sequence.

Point #2:- Some of the pulldowns in the PhytoAFP platform (http://bioinformatics.cimap.res.in/sharma/PhytoAFP) was not available, especially no database can be accessed.

Response:  We agree with the reviewer's opinion and have corrected and revised the web page of the PhytoAFP web server. We have improved the download options on the PhytoAFP webpage. All the datasets are now available at the webpage of PhytoAFP webserver.

Point #3: - In order to well predict the therapeutic anti-fungal peptides, the authors should carefully choose the training database with the known therapeutic anti-fungal peptides. The authors should carefully describe their criterions of choosing the sequences in the training database.

Response:

Our dataset consists of two types of datasets, positive(anti-fungal) and negative(non-antifungal). We have extracted 510 experimentally validated unique plant based anti-fungal peptides (PhytoAFP) as our main dataset. These were manually curated from PlantAFP database for SVM learning. We have excluded modified or non-natural amino acids which can be used in peptide sequences from our studies. We collected random peptides (510) from UniProt database and considered them as negative dataset (Non-PhytoAFP). Thus, the PhytoAFP main dataset contains 510 PhytoAFP and 510 Non- PhytoAFP.

Point #4:- The authors should also carefully describe the logics of their algorithm and show the efficacy of their prediction. In other words, the test database should be built and the predicted anti-fungal “leads” should be proved experimentally.

Response:  In the present study, we developed models for discriminating antifungal and non-antifungal peptides using a highly successful machine learning technique. We developed SVM models using SVMlight package. Support Vector Machine (SVM) is a supervised machine learning algorithm which can be used for both classification and regression challenges. This package is powerful as well as user-friendly where we can adjust the parameters and kernel functions like Linear, Polynomial, RBF and Sigmoid. Earlier, the Author (Atul Tyagi) developed the computational algorithms for designing therapeutic peptides. The author (Atul Tyagi) has played a major role in development of webservers such as AntiCP[21], TumorHPD[23], and CellPPD[25], to predict and design anticancer peptides, tumor homing peptides, and cell-penetrating peptides, respectively.

Various features that include amino acid composition, -di -tri peptide, split amino acid composition and binary profile of pattern were used as input features. In order to test the performance of the models, a five-fold cross-validation technique is used. Four sets were used for training and remaining one in used for testing. The whole process is repeated five times. Threshold dependent parameter based on Accuracy and the MCC measurements revealed Di- and Tri- peptide SVM models were selected as best models. The performance of the selected models was evaluated on independent dataset. To validate our model, we evaluated the performances of our best models (Di and Tri-peptide composition) on an independent dataset.

In the case of Tri-peptide composition-based model, we found the maximum accuracy of 94.40% and MCC of 0.89 for the training dataset and accuracy of 90.05% and MCC value 80% for the validation dataset. Similarly, Di-peptide composition-based model, achieved accuracy 94.28% with MCC 0.89 for the training dataset and accuracy of 91% and MCC value 0.82% for the validation dataset and suggesting that our models are useful for further studies.

Point #5:- The manuscript should be carefully proof-read.

Response: We have rewritten most of the contents under different sections keeping the original subject matter intact. We have carefully carried out the double proof- reading process in order to be fully satisfied about the manuscript.

Reviewer 3 Report

The submitted manuscript “PhytoAFP: In silico approaches for designing plant-derived antifungal peptides” by Tyagi et al developed an in silico web server, PhytoAFP, to predict and design antifungal peptides of plant origin. This server can calculate physicochemical properties, including charge, hydrophobicity and pI. It is a useful tool for the development of new antifungal peptides against emerging infectious diseases. However, there is no conclusion of this manuscript to summarise their findings. Therefore, I recommend this manuscript be published after major revision.

There are additional minor comment,

page 2, line 46 and 55, they briefly mentioned the definition of AMPs. However, they should also refer to the recent excellent revs (such as Chem. Soc. Rev., 2021,50, 4932-4973 https://doi.org/10.1039/D0CS01026J, Lancet Infect Dis 2020; 20: e216–30, https://doi.org/10.1016/S1473-3099(20)30327-3))

Author Response

PhytoAFP webserver comments point-by-point response to Ref.: Ms. No. 1241099

Thank you very much for yourcomments, and suggestions for helping us improve the manuscript. We have incorporated all your answers to your comments and suggestions into our revised manuscript.We have rewritten most of the content under different sections, keeping the original subject matter intact in our manuscript. Changes are highlighted in blue color. We are hereby addressing the responses on a point-by-point basis.

Reviewer #3: PhytoAFP webserver comment:

The submitted manuscript “PhytoAFP: In silico approaches for designing plant-derived antifungal peptides” by Tyagi et al developed an in-silico web server, PhytoAFP, to predict and design antifungal peptides of plant origin. This server can calculate physicochemical properties, including charge, hydrophobicity and pI. It is a useful tool for the development of new antifungal peptides against emerging infectious diseases. However, there is no conclusion of this manuscript to summarize their findings. Therefore, I recommend this manuscript be published after major revision.

Point #1: -- page 2, line 46 and 55, they briefly mentioned the definition of AMPs. However, they should also refer to the recent excellent revs (such as Chem. Soc. Rev., 2021,50, 4932-4973 https://doi.org/10.1039/D0CS01026J, Lancet Infect Dis 2020; 20: e216–30, https://doi.org/10.1016/S1473-3099(20)30327-3))

Response: We took the opportunity to provide detailed “Discussion” so that most of the contents are clear and provide valuable information to the readers pertaining to the scientific community. We had not incorporated the “Conclusion” section as it was not mandatory to provide according to the journal guidelines; however, we have included the Conclusion section in the updated manuscript suggested by the reviewer [see line number 317-323].

The definition of AMPs is clearly described in the “Introduction” section.  The current references suggested by the Reviewer is simultaneously updated in the manuscript and reference Section [see line number 60-63].

Round 2

Reviewer 1 Report

The suggestions made previously were addressed by the authors. The modification of the discussion proved to be beneficial for the quality of the manuscript.

I have no further questions regarding this manuscript.

Author Response

Response to Reviewer 1 Comment

Point 1: The suggestions made previously were addressed by the authors. The modification of the discussion proved to be beneficial for the quality of the manuscript. I have no further questions regarding this manuscript.

 Response 1:  Thank you very much for the suggestions and comments of the reviewer and the accepted our manuscript in the Journal of Antibiotics.

Reviewer 2 Report

The authors have answered my questions and comments and improved both the results and discussion parts of the manuscript.  I agree to accept the paper published in Antibiotics.  But, please check the grammar and spelling by a professional English editor.

Author Response

Response to Reviewer 2 Comments

Point 1: The authors have answered my questions and comments and improved both the results and discussion parts of the manuscript.  I agree to accept the paper published in Antibiotics.  But please check the grammar and spelling by a professional English editor. 

Response 1:  Thank you very much for the suggestions and comments of the reviewer and the accepted our manuscript in the Journal of Antibiotics. We have carefully carried out the double proofreading process and hope to be completely satisfied with the manuscript.

Reviewer 3 Report

The authors have addressed my comments.

Author Response

Response to Reviewer 3 Comments

Point 1: The authors have addressed my comments

Response 1:  Thank you very much for the suggestions and comments of the reviewer and the accepted our manuscript in the Journal of Antibiotics.
